# Drug Use Control Perception and Strategies in General and Clinical Population in a Spanish City

**DOI:** 10.3390/ijerph18179189

**Published:** 2021-08-31

**Authors:** Salvador Amigó, Angela Beleña

**Affiliations:** Department of Personality, Evaluation and Psychological Treatments, Faculty of Psychology, University of Valencia, 46010 Valencia, Spain; mangeles.belena@uv.es

**Keywords:** drugs, perception of controlling drug use, drug use control strategies, moderate drug use, risk and harm reduction approach

## Abstract

Background: This article evaluates the perception of drug use control and strategies in Valencia City (Spain) in a general and clinical population, in two independent studies. Material and Methods: 1071 people participated. In Study 1 (*n* = 924) the entire sample came from the general population (GP), and in Study 2 (*n* = 147), 68 were drug users being treated in an Addictive Behavior Unit (ABU), and 79 people of the GP. The drug use control perception and strategies in both subgroups were compared. The participants filled in the Drug Use Strategies Scale and a Drug Use Survey. Results: A high level of perception of drug control in the GP was obtained (72.7% in Study 1 and 67.5% in Study 2), and 32.5% in the ABU subgroup. People in the GP and drug users in treatment differ in some control strategies. A predictive profile of the perception of control was obtained for Study 2. Conclusion: The high degree of perception of controlling drug use in the GP, and partially in drug users being treated, and the specific control strategies reported suggests that moderate use and drug control strategies are a great value alternative to bear in mind compared to abstinence.

## 1. Introduction

There is growing evidence that controlled drug use is more the rule than the exception, and that personal control is relevant even in cases of addiction [1]. Zinberg in [2] argues that controlled drug use is the result of not only the social context in which drugs are taken but also of attitudes to drugs, learned self-control skills and personality factors. Thus the social context in which drugs are used favours controlled use, and encourages social rituals and sanctions of unsuitable or high-risk drug use.

Community studies conducted with large samples of heavy cocaine drug users and other drugs around the world have shown that controlled cocaine use is very frequent, which refutes the myth of the “inevitability of the climb from drug use to addiction” [3,4,5,6,7,8,9,10,11,12]. One of these studies defines “controlled use” as “a self-controlled use pattern that does not derive from any significant social dysfunction in, for example, any alteration in skills to assume the roles and responsibilities in daily living” [11] (p. 33).

Several authors have also described the strategies that are most widely used by users of all kinds of drugs [13,14,15,16]. For instance, it was verified that cocaine users can enhance their drug use control by following certain rules [17], such as: (1) do not use drugs on a daily basis; (2) set rules as to how and when to use drugs; (3) do not commercialise drugs, etc. Vadhan et al. in [18] checked that heavy cocaine users were able to make a decision about not using drugs if they were offered an alternative, such as money. Other studies conducted in the San Francisco Bay Area and in Sweden with frequent users of cannabis and other drugs have verified that drug users also adopt rational criteria to control their drug use, by acting normally in their day-to-day lives, and by reducing possible risks and harm caused by drugs [19,20].

Another research line consisted in creating and applying scales to assess drug control strategies. The Protective Behavioral Strategies Scale (PBSS) is a 20-item scale that presents alcohol management strategies. It was created using information about the alcohol management strategies found in the scientific literature, on reports about alcohol users, particularly university students, and on experts in this study area. In line with the perspective of reducing risks and harm caused by drugs, this scale evaluates adopting certain strategies to reduce the harm caused by drinking alcohol [21,22]. A high PBSS score is associated with less alcohol use and fewer problems related to alcohol (e.g., [21,23,24,25]).

A study into drug use strategies [26] asked young people from Valencia (Spain) if they thought it was possible to control drug use by adopting certain strategies and, if they thought it was, which strategies did they think were the most useful ones. This was performed using the Drug Use Strategies Scale (DUSS). Of the 724 youths who answered the question, 471 (65%) responded YES and 253 (35%) NO. Of the 17 proposed strategies, the most highly valued were: Strategy 1 (“Do not use drugs on a daily basis”), 4 (“Do not use drugs to sort out my problems or faults”), 8 (“Do not use drugs to escape from reality”), 12 (“Reduce the amount of drug”), 14 (“Keep my mind occupied and do other things when I feel like using drugs”) and 17 (“Think about the negative personal and health consequences”).

Fewer studies are available on drug control strategies adopted by users being treated. For instance, Lin and Zhang in [27] conducted a study in a rehabilitation center in Shanghai, where patients participated in in-depth interviews about their synthetic drug use. Users reported that they adopted control strategies, e.g., selecting and using drugs according to their own experience and in line with friends’ suggestions; controlling doses; limiting doses by spacing them in time and avoiding coming into contact with heavy drug users; trusting in the group’s rules to avoid overdoses; enjoying the positive effects of the drug more.

This article aimed to confirm the first results obtained with the DUSS and to also verify if drug users being treated are in favour of adopting drug use strategies or not compared with a sample of users from the general population. 

Our main hypothesis is that the results on the perception of drug use control that have been obtained in numerous cities around the world, as we mentioned previously, and that has been obtained in a sample of first-year university students in the city of Valencia (Spain) [26] will be replicated in this study with a broader and more diversified sample. The second hypothesis refers to the fact that we will find a percentage of drug users with a lower perception of drug use control than in the general population, but a significant one, in such a way that the harm and risk reduction approach as an alternative to abstinence in these drug users must be considered.

Knowledge of the perception of controlling drug use and preferred control strategies provides relevant information about the resources available to drug users to handle drug use and will offer valuable information for professionals to design efficient prevention and/or intervention programs.

## 2. Materials and Methods

### 2.1. Participants

This article presents two studies: 924 volunteers participated in Study 1, and 147 volunteers participated in Study 2. The whole sample was composed of 1071 volunteers, all of them from Valencia city (Spain).

In Study 2 two samples were compared. The first sample was made up of 68 drug users being treated in the Addictive Behavior Unit (ABU). The second sample was formed by 79 people from the general population (GP), that they were not drugged users or that they were not or had not been treated for drug addiction. 

Table 1 offers data on age, level of education and gender, which correspond to each group in both studies. In order to present the collected results, in this table and the following ones, we will present the materials and methods for the two studies together.

In Study 1 we can see a higher percentage of women (61.4%) than men (38.6%), an average age of 28.10 (SD = 11.18), with almost half being university students (46.5%).

In Study 2 the sample included a higher percentage of males than females for both the ABU (67.6% and 32.4%, respectively) and GP (53.2% and 46.8%, respectively) subgroups. Ages were somewhat older in the ABU than in the GP. For the level of education, we can clearly see a higher level of education in the GP subgroup, with 53.2% as opposed to 8.8% of university students in the ABU subgroup.

So, the participants in Study 1 are younger than those in Study 2, and also have a higher percentage of women and university students. Due to the fact that the *n* is quite different in both studies (*n* = 924 in Study 1 and *n* = 147 in Study 2), and the participants in Study 1 totally came from General Population, the comparisons between the general and clinical samples will only be carried out in Study 2, that was planned with that objective (*n* = 79 for GP subgroup and *n* = 68 for ABU subgroup).

### 2.2. Instruments

Drug Use Strategies Scale (DUSS) [26]. It includes several drug use control strategies, which must be answered by those who positively responded to this question: “Some people say that they control drug use because they do certain things that moderate drug use or reduce risks. Do you believe that it is possible to moderate, control or reduce drug use in general (those considered “hard” and “soft”, legal or illegal) by doing certain things or employing certain strategies?”. DUSS is a 17 item scale on yes/no response that assesses drug use control strategies in different areas: restricting drug use (e.g., “Not taken on a daily basis”), not using drugs to sort out personal problems or shortcomings (e.g., “Do not use drugs to overcome my problems or faults”), reducing the amount of drug (e.g., “Propose a limited quantity for each day”) or looking for alternatives (e.g., “Alternate drug use with other activities (walking, reading, etc.”)).Drugs Survey following the European Monitoring Centre for Drugs and Drug Addiction (EMCDDA) criteria [28]. It is a brief self-report questionnaire, which measures the frequency of drugs use (such as cannabis, alcohol, tobacco, cocaine, MDMA, sedatives, hallucinogens and amphetamines). People have to answer questions such as: Sometime in your life; How often in your life; How often in the last 12 months; and How often in the last month.

### 2.3. Inclusion Criteria for Participation and Obtaining the Sample

The participants in the ABU subgroup had to fulfil drug use/drug addiction diagnosis criteria and be treated. The other subgroup (the GP) was obtained by the “snowball” method from the GP. None of these participants had ever been on treatment and did not fulfil the drug use/drug addiction diagnosis criteria. The same criteria were used in Study 1. In these cases, students from the Faculty of Psychology and the Polytechnic University of Valencia filled out the questionnaires and were asked to find other people to fill them out at the same time. They were instructed to look for both drug users and abstainers, both men and women, and people of different ages. The questionnaires were distributed both by hand and via email, but always individually and personally.

Study 1 was a previous study and the sample came entirely from university students of the Polytechnic University of Valencia. Regarding the GP and ABU samples of study 2, the greatest homogeneity was sought in terms of gender and age, but homogeneity was not possible at the level of studies, since more than 50 percent of the members of the GP group are university students (greater access with the snowball) and less than 10 percent are in the ABU group. The ABU group sample came precisely from Addictive Behavior Units and all users were included at the request of the center’s psychologists.

All subjects gave their informed consent for inclusion before they participated in the study. The study was conducted in accordance with the Declaration of Helsinki, and the protocol was approved by the Ethics Committee of the University of Valencia (Spain) in 2017 (Project identification code: H1484824011097).

## 3. Results

Data were analysed using IBM Corp. Released 2015. IBM SPSS Statistics for Windows, Version 23.0. IBM Corp (Armonk, NY, USA).

Table 2 shows the numbers and percentages of the participants who had used drugs sometime in their life, such as cannabis, ecstasy, cocaine, amphetamines and hallucinogens, for all groups. Study 2 also provides the Chi-square test results, which compare the frequencies of both subgroups. In all the analyses, 0 boxes indicate an expected frequency below 5, thus applying the Chi-square test was appropriate.

Participants in Study 1 (who all come from the general population) have somewhat higher percentages of drug use sometime in their life than those in the GP subgroup in Study 2, and for all drugs, but lower than those of the ABU subgroup.

For Study 2, the use of drugs some time in their life was higher for the ABU subgroup than for the GP subgroup for all the drugs. The Chi-square test was significant for all drugs. In both subgroups, the highest percentage corresponded to cannabis.

Table 3 offers drug use data (levels of drug use in one’s life, in the last 12 months and in the last month, as percentages) for all study groups.

We can see how the participants in Study 1 have used drugs more recently (in the last year and in the last month) than the participants in the GP subgroup of Study 2, especially cannabis, ecstasy and cocaine.

For Study 2 we can see, drug use in one’s life (>30 times) is clearly higher for the ABU subgroup than for the GP subgroup, and for all drugs, although this difference is somewhat smaller for cannabis. The use of drugs in the last 12 months is also higher for the ABU subgroup members compared to the GP ones. However, their drug uses in the last month do not differ that much as the ABU group members are being treated.

Table 4 offers the percentage of participants who consider that it is possible to adopt certain strategies to control drug use.

The highest percentage of perception of drug use control was obtained in Study 1 (72.7%).

For the Study 2 (*n* = 147) it was 56.5%. When we compared both study subgroups, the percentage of participants who responded YES in the GP group doubled this response in the ABU group (67.5% and 32.5%, respectively). The Chi-square test and the contingency coefficient were significant.

The YES / No response ratio in the GP group is: 53/26 = 2.0384; and in the ABU group is: 27/41 = 0.6585. The odds ratio is: 2.0384/0.6585 = 3.095. This can be interpreted as being three 3.095 times more likely to find the YES answer in the GP group than in the ABU group.

It is worth stressing that although the percentage of ABU subgroup members was lower than that which corresponded to the GP subgroup, it was striking that 32.5% thought that it was possible to control drug use by adopting certain strategies.

Next, a binary logistic regression analysis was proposed to determine which variables predicted perceived control. To this end, perceived control was considered a dependent variable, while group and gender are dichotomous variables that contain precisely two values (drug users being treated versus general population for group, and female and male for gender), age, level of education, and having used drugs some time in one’s life were the independent variables.

For the level of education variable, three dummy variables were created. As most of the participants had used several of the drugs contemplated herein more than once in their life, these variables could correlate. Therefore, analyses were previously carried out of tolerance and VIF (variance inflation factors) to rule out the effect of collinearity. The level of tolerance of all the independent variables was above 0.10, and the VIF values were under 10, which indicates no collinearity among the independent variables.

Then a hierarchical binary logistic regression analysis was performed. A stepwise selection method of variables was used, the Forward Selection (Conditional). As the independent variables, the first block included the epidemiological variables, while using drugs some time in one’s life was included in the second block.

In block 0, the overall statistics (*p* < 0.0005) indicated that there was a relation between the independent variables and the dependent variable. The analysis was done in two steps, with a Nagelkerke R^2^ of 0.277 and 0.324, respectively. The final analysis results are shown in Table 5.

As the education variable was not significant, it was not necessary to include the dichotomous variables that derived from the categorical variable.

It is possible to interpret the estimated values of B (obtained from the marginal frequencies of the dependent variable) in Table 5 as coefficients of the logistic equation resulting from the logit transformation of the original logistic function and this allows to express the logit transformation as a linear combination of effects. It allows us to work with a linear model. The value of this coefficient indicates how the logit of perception of control (the linear forecast of the logistic equation) changes for each unit that the value of the covariate increases. This value is expressed on a logarithmic scale. Returning it to its natural scale gives exp (B) which is the odds rate. With all this, and taking into account the sign, we can make the following interpretation of the results: the logit of perception of control was: (a) 1.12 times higher for males; (b) 1.89 higher for the GP group; (c) 0.40 higher for older people; (d) 1.11 higher for those who had used cannabis sometime in their life. The variables that better explain the perception of control are those with higher Exp(B): gender and having used cannabis once in their life.

Next the participants who answered YES to the question about possibly controlling drugs by adopting certain strategies were selected (*n* = 672 for Study 1 and *n* = 83 for Study 2). For Study 2, 27 belonged to the ABU subgroup and 56 to the GP subgroup. These were the participants who later indicated which strategies were on the list of 17 strategies they thought were useful. Table 6 indicates the frequencies and percentages that corresponded to both studies, as well as the results of the Chi-square test and the contingency coefficient for the ABU-GP comparisons of Study 2.

It is interesting to note that the percentages of the general population both for Study 1 and 2 are very similar, taking into account the difference in the sample size (*n* = 672 and *n* = 56, respectively). 

When comparing both subgroups of Study 2, we can firstly see that differences in percentages were found according to relative frequencies. 

To better analyse the obtained results, we opted to divide the percentages into three categories: low-frequency strategies, medium-frequency strategies, high-frequency strategies. We considered that those who obtained a percentage of 33% or lower were low-frequency strategies, those with a percentage between 33% and 66% were medium-frequency strategies, and those with a percentage above 66% were high-frequency strategies. As a result, although the GP subgroup members indicated more high-frequency strategies (12) than the ABU subgroup members (9), the number of strategies indicated by the ABU subgroup is far from negligible, and they also indicate more medium-frequency strategies (8 vs. 4 in the GP group). 

The comparison of the frequencies and percentages between both groups provided highly relevant, but incomplete, information as expected frequencies were not taken into account. This was why a Chi-square test was carried out. In seven strategies significant results were obtained with both the Chi-square test and contingency coefficient. The GP subgroup members indicated these strategies with a significantly higher frequency than the ABU subgroup did. It is worth stressing these seven strategies: Strategy 1(Not taken on a daily basis); Strategy 3 (Do not sell drugs); Strategy 4 (Do not use drugs to overcome my problems or faults); Strategy 6 (Alternate drug use with other activities (walking, reading, etc.); Strategy 8 (Do not use drugs to escape from reality); Strategy 15 (Do not go out with people who use drugs or go to places where they are taken); Strategy 16 (Keep only the amount of drug to take for one day; do not store at home).

When we examined these seven strategies, we saw that apart from “Do not sell drugs” (Strategy 3), where we would expect GP subgroup members to respond much more favourably than ABU subgroup members, we could group the other six strategies into two groups: (1) restricting quantity, days when the drug is taken, and places and people, as well as seeking alternatives (Strategies 1, 6, 15 and 16); (2) not using drugs to sort out personal problems or to escape from reality (Strategies 4 and 8). For the other strategies, the proportions of the responses between the two study groups were similar.

## 4. Discussion

This study provides some noteworthy results. In the first place, Study 1 replicates the results obtained previously in relation to the perception of control of drug use [26]. In the present study, with a larger and more diverse sample, the favourable view of using drug use control strategies was even higher (*n* = 72.7) than in the previous study (*n* = 65%).

For Study 2, of all the participants (*n* = 147), slightly over half (56%) considered that it was possible to control drug use by adopting certain strategies. We named this attitude “perception of control”. In the GP subgroup, 67.5% agreed (two-thirds of this group), which almost completely coincides with the first results obtained by [26], when 65% responded YES.

One novel aspect of the present study was that 32.5% of the drug users being treated in an ABU also answered YES. Although this percentage was slightly less than half of those who answered YES in the GP group, it is interesting to verify that one-third of those being treated for drug addiction considered it possible to control drugs.

The next significant result obtained in this study was to obtain a characteristic profile, related with higher perceived control, with the following variables: (1) belonging to the GP group; (2) being male; (3) being older; (4) having used cannabis sometime in one’s life. 

On the other hand, the strategies with the highest scores obtained for the general population in both Study 1 and 2 (1, 4, 8, 12, 14 and 17 strategies) correspond perfectly with those obtained in the previous study [26]. This is an interesting and important result, since adding these samples (*n* = 1727; since the general population of the research presented here is *n* = 924 + 79, and that of the previous study cited is *n* = 724), we can say that we identified the strategies most used by the general population, at least in the Mediterranean city of Valencia.

Finally, we can see that significant differences (Chi-square) were found between the two subgroups of Study 2 for seven of the 17 evaluated strategies and that the GP subgroup members pointed out these strategies more frequently. In short, we can state that the GP subgroup members considered it necessary to restrict drug use by bearing in mind places and people, and they also rejected using drugs to sort out personal problems or shortcomings. This indicates that perceived control in the GP is related to various kinds of control strategies, which are richer and ampler. The drug users being treated did not sufficiently consider the possibilities of strategically restricting drug use; perhaps use drugs as an emotional coping strategy. It was previously verified for coping that ABU group members use the drug resource and cognitive escape more, whereas GP members use cognitive coping with problems more [29,30].

At this point, it would be most interesting to contemplate the implications that the results indicate to intervene in drug matters.

Firstly, we can state that GP members are more predisposed than drug users being treated to follow certain drug control use strategies that can be proposed according to a drug abuse prevention program, and not from a total abstinence objective, but by moderating drug use. This would work particularly well for those who have taken cannabis and are males and older individuals.

However, does this mean that programs to reduce or moderate drug use, as alternatives to abstinence, would not be useful for the other users (GP members and those being treated for their addiction)? We believe proposing the two alternatives would be most useful for the other drug users: (1) abstinence and zero drug use, or; (2) moderating drug use by following drug use control strategies. Similar strategies have been reported in a study into cocaine drug addicts, which showed that the coping strategies which related more to reduced drug use were “thinking about the negative consequences”, “distractions”, “alternative conducts”, etc. [31].

This is true because we need to bear in mind that, as previously mentioned, one-third of drug users being treated consider that controlled drug use is possible; moreover, the centers where they are being treated apply intervention programs to achieve abstinence. It is quite possible that if they were offered other alternatives, such as moderate drug use, a higher percentage of drug addicts being treated would agree. However, this is a matter for future research.

With the drug users being treated, it will first be necessary to insist on setting rules for drug use to strategically restrict the number of drugs and the conditions under which they are taken (places, company, etc.); secondly, it would be necessary to teach them coping strategies for their problems, which would be alternatives to using drugs. In some treatments that promote moderate drug use, such as the Brief Alcohol Screening and Intervention for College Students—BASICS program [32], drug users are encouraged to adopt coping strategies to reduce risks [24,33,34,35,36].

Training in using protective behavioural strategies for alcohol use has offered contradictory results. Although poorly satisfactory results have been obtained when applying for only personalised and brief strategy training programs, they obtain clearly better results when they are prolonged in time or are extended with motivational interviews, social norms and group techniques [37,38,39].

Hence it is worth considering the potential use of strategies to control drug use as a fundamental component of drug use prevention programs and to also treat drug users. There is some evidence that using drugs such as heroin can be controlled to a certain extent [40]. Treatment programs that have been designed to apply control strategies with alcoholics and cocaine addicts have managed to reduce drug/alcohol use and relapses (see [41] for a review). There is evidence to suggest that training programs for behavior control have been useful for opioid [42] and cannabis users [43].

A pilot study has tested the efficiency of an online program to reduce drug use by employing the 17 drug use control strategies by means of the DUSS [44]. The 14 drug users who completed treatment were able to significantly reduce their drug use by increasingly applying the proposed strategies.

This study has its limitations. It is necessary to increase the number of drug users being treated in this study, including drug users samples who are not being treated and others who have finished their rehabilitation programs. It would also be interesting to relate the perception of drug use control with other variables, such as stress coping strategies, personality, attitudes to drugs, etc., which would provide a better understanding of the drug control perception. Evidence was found for a relationship between personality variables, such as sensation-seeking and impulsiveness, and perception of control, and also with several of the 17 DUSS strategies [45,46,47,48]. The predictive profile of the favorable perception of drug control obtained in this study is a step in that direction. On the other hand, the answer to the question about the perception of control is necessarily subjective. Although the replication of the percentages in different studies and the type of strategies chosen seem to indicate an acceptable degree of sincerity, it will be future research in the application of drug abuse prevention programs using the proposed strategies that can confirm the sincerity of their answers. As indicated above [44], that has already begun to be verified.

## 5. Conclusions

Although these results are preliminary, the intervention to reduce drug use based on control strategies is an open research field. In such a hard and complex area as drug use and drug addiction, being able to rely on valid alternatives to abstinence programs can be extremely useful. This article is a step forward in this direction.

In this sense, the initial hypotheses have been confirmed. Therefore, the result obtained in a previous study in the Spanish city of Valencia (65% of the general population with a favourable view of the perception of drug use control and the use of strategies for it) was replicated with an even larger sample, with an even higher percentage of 72.2%.

On the other hand, we found a percentage of drug users with a lower perception of drug use control than in the general population, but a significant one (32.5%) and we were able to identify drug use control strategies in which drug users in treatment and the general population coincide and differ.

All those results provide information about the perception of control and evaluate strategies for moderate drug use, thus offering valuable material for developing drug use risk and harm reduction programs.

## Figures and Tables

**Table 1 ijerph-18-09189-t001:** Gender, age, and level of education of the participants in this study (*n* = 1.071).

Demographic Variables		Study 1 (*n* = 924)			Study 2 (*n* = 147)					
					ABU (*n* = 68)			GP (*n* = 79)		
**Gender**	Male	357 (38.6)			46 (67.6)			42 (53.2)		
	Female	567 (61.4)			22 (32.4)			37 (46.8)		
**Studies**	No qualifications	53 (5.7)			5 (7.4)			0		
	Secondary education	273 (29.5)			38 (55.9)			15 (19)		
	Vocational training	168 (18.2)			19 (27.9)			22 (27.8)		
	University	430 (46.5)			6 (8.8)			42 (53.2)		
**AGE**		**Mean**	**SD**	**R**	**Mean**	**SD**	**R**	**Mean**	**SD**	**R**
		28.10	11.18	55	38.99	9.77	41	32.99	11.87	40

ABU: Addictive Behavior Unit; GP: General Population; *R* = age range. Percentages are in brackets.

**Table 2 ijerph-18-09189-t002:** Taken drugs sometime in their life for the three groups of this study.

Drugs	Sometime in Their Life				
	Study 1	Study 2			
		TOTAL	ABU	GP	χ^2^ (Sig.)
Cannabis	339 (69.2)	100 (68)	54 (79.4)	46 (58.2)	7.54 (0.005)
Ecstasy	167 (18.1)	43 (29.3)	34 (50)	9 (11.4)	26.32 (0.000)
Cocaine	217 (23.5)	67 (45.6)	55 (80.9)	12 (15.2)	63.58 (0.000)
Amphetamine	121 (13.1)	39 (26.5)	35 (51.5)	4 (5.1)	40.37 (0.000)
Hallucinogens	178 (19.3)	47 (32)	34 (50)	13 (16.5)	19.90 (0.000)

ABU: Addictive Behavior Unit; GP: General Population. Percentages are in brackets.

**Table 3 ijerph-18-09189-t003:** Levels of drug use in one’s life, in the last 12 months and in the last month.

Drugs	Studies		How Often in Your Life			How Often in the Last 12 Months				How Often in the Last Month			
			1–5	6–30	>30	0	1–5	6–30	>30	0	1–5	6–30	>30
Cannabis	Study 1		34.4	19.2	45.9	45	21.1	12.2	21.4	64.3	16.4	7	12.1
	Study 2	ABU	18.5	13	68.5	40.7	16.7	29.6	59.3	66.7	13	5.6	14.8
		GP	28.3	28.3	43.5	58.7	26.1	8.7	6.5	80.4	10.9	8.7	0
Ecstasy	Study 1		50.9	29.3	19.8	59.3	29.9	7.8	3	88	9.6	1.8	0.6
	STUDY 2	ABU	38.2	23.5	38.2	88.2	8.8	2.9	0	97.1	2.9	0	0
		GP	77.8	11.1	11.1	55.6	44.4	0	0	88.9	11.1	0	0
Cocaine	Study 1		37.8	24.4	37.3	48.4	29.5	14.3	7.8	79.2	13.1	3.7	0.9
	Study 2	ABU	5.5	7.3	87.3	45.5	16.4	9.1	29.1	78.2	18.2	1.8	1.8
		GP	50	25	25	75	16.7	8.3	0	91.7	8.3	0	0
Amphetamine	Study 1		47.1	29.8	23.1	59.5	24	12.4	4.1	81.8	14.9	3.3	0
	Study 2	ABU	25.7	22.9	51.4	82.9	14.3	0	2.9	94.3	5.7	0	0
		GP	75	0	25	50	25	25	0	50	50	0	0
Hallucinogen	Study 1		66.3	25.3	8.4	67.4	25.3	7.3	0	87.1	12.4	0.6	0
	Study 2	ABU	44.1	23.5	32.4	94.1	5.9	0	0	100	0	0	0
		GP	84.6	7.7	7.7	92.3	7.7	0	0	100	0	0	0

ABU: Addictive Behavior Unit. GP: General Population.

**Table 4 ijerph-18-09189-t004:** Perception of drug use control for the whole sample.

Control Strategies	Study 1	Study 2				
		Total	ABU	GP	χ^2^ (Sig.)	Contingency Coefficient (Sig.)
YES	672 (72.7)	83 (56.5)	27 (32.5)	56 (67.5)	14.45 (0.000)	0.299 (0.000)
NO	252 (27.3)	64 (43.5)	41 (64.1)	23 (35.9)		

Percentages are in brackets. ABU: Addictive Behavior Unit; GP: General Population.

**Table 5 ijerph-18-09189-t005:** Results of the hierarchical binary logistic regression analysis.

Variables	B	Wald	Sig.	Exp(B)
Group	−1.891	9.061	0.003	0.151
Age	−0.040	4.484	0.034	0.961
Gender	1.123	6.230	0.013	3.075
Education	−0.079	2.88	0.409	0.877
Cannabis	1.118	4.423	0.035	3.058
Ecstasy	−0.056	0.008	0.928	0.945
Cocaine	0.076	0.012	0.914	1.079
Amphetamine	−0.763	1.385	0.239	0.466
Hallucinogen	−0.037	0.003	0.956	0.964

**Table 6 ijerph-18-09189-t006:** Favourable views (in percentages) of using some of 17 drug use control strategies for all groups.

	Control Strategies	Study 1 (*n* = 672)	Study 2			
			ABU (*n* = 27)	GP (*n* = 56)	χ^2^ (Sig.)	Contingency Coefficient
1	Not taken on a daily basis	620 (92.3)	20 (74.1)	52 (92.9)	5.59 (0.018)	0.25
2	Only taken in certain places or with certain people	408 (60.7)	15 (55.6)	35 (63.6)	0.49 (0.48)	0.07
3	Do not sell drugs	526 (78.3)	18 (66.7)	49 (87.5)	5.08 (0.024)	0.24
4	Do not use drugs to overcome my problems or faults	603 (89.7)	20 (74.1)	53 (94.6)	7.27 (0.007)	0.28
5	Only take orally or by snorting (if cocaine is used)	154 (22.9)	10 (37)	14 (25)	1.28 (0.257)	0.12
6	Alternate drug use with other activities (walking, reading, etc.)	496 (73.8)	18 (66.7)	49 (87.5)	5.08 (0.024)	0.24
7	Subordinating my obligations to drug use (getting up early to go to work, even with a hangover)	484 (72)	20 (74.1)	41 (73.2)	0.007 (0.934)	0.009
8	Do not use drugs to escape from reality	570 (84.8)	19 (70.4)	49 (89.1)	4.48 (0.034)	0.22
9	Consume only soft drugs such as cannabis	235 (35)	14 (51.9)	25 (45.5)	0.297 (0.586)	0.06
10	Do not use drugs alone, but with others	312 (46.4)	13 (48.1)	30 (55.6)	0.39 (0.529)	0.07
11	Do not take drugs at home, always somewhere else	257 (38.2)	12 (44.4)	28 (50)	0.225 (0.635)	0.05
12	Reduce the amount of drugs	596 (88.7)	22 (81.5)	52 (92.9)	2.43 (0.118)	0.16
13	Propose a limited quantity for each day	470 (70)	19 (70.4)	41 (74.5)	0.161 (0.688)	0.04
14	Keep my mind occupied when I want to take drugs	635 (94.5)	25 (92.6)	54 (96.4)	0.58 (0.445)	0.08
15	Do not go out with people who use drugs or go to places where they are taken	562 (83.6)	21 (77.8)	55 (98.2)	9.85 (0.002)	0.32
16	Keep only the amount of drug to take for one day; do not store at home	463 (68.9)	15 (55.6)	45 (81.8)	6.36 (0.012)	0.26
17	Think about the negative personal and health consequences	622 (99.6)	26 (96.3)	54 (96.4)	0.001 (0.976)	0.003

Chi-square and contingency coefficient for variables ABU (*n* = 27) and GP (*n* = 56). ABU: Addictive Behavior Unit; GP: General Population.

## Data Availability

The data presented in this study are available on request from the corresponding author.

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
