# Peer review of "Drug Use Control Perception and Strategies in General and Clinical Population in a Spanish City"

_ijerph, 2021, doi:10.3390/ijerph18179189_

Round 1

Reviewer 1 Report

I have read the manuscript with interest, as it covers a topic often overlooked in the mainstream literature on drug policies and drug prevention: perception of control and control strategies in relation with drug use. The manuscript reports two studies, one of them with a relatively large community sample, and a second one comparing clinical and community-dwelling samples. The study of two types of samples is a clear strenght, which enrich the scope of the research, and allows interesting derivations for prevention and harm reduction.

I will make some minor suggestions to improve readability and replicability.

  1. The introduction nicely reports on the relevant background. Nevertheless, I think the text is formally too fragmented, and this undermines the reading flow. I suggest the authors to integrate/join the short paragraphs which deal with close contents: e.g., lines 34 and 35, lines 42 and 43.
  2. Be more specific in describing instruments. Even if the strategies covered by the DUSS are outlined later in the manuscript, the reader will like to see at least some examples when the scale is introduced in the "Instruments" section. Response options could be also reported.
  3. Some more information could be added to describe the procedures. How was the "snowballing" conducted? Starting from university students? through acquaintances? through social media?
  4. There is not much information on the selection of the GP subsample in Study 2.  GP subsample in Study 2 differs substantially from GP sample in Study 1, which would be unexpected if the recruiting procedure was exactly the same for both studies. I wonder whether some specific efforts were made in the recruiting process so that GP subsample (Study 2) was comparable, in terms of age and gender, to the ABU subsample.
  5. As far as I could see, age Mean and SD are appropriately reported, but the range of ages in each group would be also informative
  6. Some minor language/typo mistakes may need to be amended: PG (line 15), "ESTUDY 1" and "ESTUDY 2" (Tables 1 and 2), "Table 4 offer" (line 168).
  7. Please note that there are some hyphened words, which may have resulted from the editing process: e.g., "inde-pendent" (page 198).

In my opinion, the manuscript offers insightful results and reflections on a relevant topic, and deserves to be known by the scientific and professional community.

Reviewer 2 Report

This study aimed to explore the effects of ‘perception of drug use control’ in addiction and identified lower degree of perception of drug use control in a group of drug user under treatment, compared with the general population. While the study being potentially important to help develop alternative treatment strategies/programs, there are some concerns/questions also need to be addressed:

The biggest concern is the subjectivity of the questionnaire. Indeed, the manuscript indicates that the addicted population showed a lower level of ‘perception of drug use control’ than the general population, however, the main conclusion of the manuscript is merely based on the subjective answer/response to the question ‘Do you believe that it is possible to control/reduce drug in general by employing certain strategies’. This response is subjective and logically not necessarily related to actual reduction/control of drug use. Therefore, the interpretation might not reflect the ‘ground truth’. For example, could it possibly be that the lower degree of the ‘perception of drug use control’ in the addiction group is based more on the actual experience and they know it didn’t work, while the relative higher degree in the general population is due to the fact that many of them never experience the actual drug withdraw/craving and simply assume ‘some strategy would work’? Even some simple follow-up that may potentially provide relationship between the observed difference in degree of perception of drug use control and the actual treatment outcome would greatly enhance the validity of the conclusion of the manuscript.

There are also some minor comments/questions to help improving the manuscript:

  1. L15: “PG”
  2. L279: why n=1721?
  3. Please check references, many references and citations are mismatched in multiple places in manuscript, for example, L67 ref#27, L333 ref#44.
  4. Some methodological details need to be provided and better organized into the ‘method’ section, for example the logistic regression on L186, what software was used? Parameters selected? Or, on L179, the ‘proportion term-based comparison’, what calculations have been done?
  5. L201, ‘sig.<.0005’, should be ‘p<.0005’.
  6. Please check unnecessary use of the hyphen in-words, for example: L181 ‘like-ly’.
